# A Novel ^89^Zr-labeled DDS Device Utilizing Human IgG Variant (scFv): “Lactosome” Nanoparticle-Based Theranostics for PET Imaging and Targeted Therapy

**DOI:** 10.3390/life11020158

**Published:** 2021-02-18

**Authors:** Melissa Siaw Han Lim, Takashi Ohtsuki, Fumiaki Takenaka, Kazuko Kobayashi, Masaru Akehi, Hirotaka Uji, Hirotsugu Kobuchi, Takanori Sasaki, Eiichi Ozeki, Eiji Matsuura

**Affiliations:** 1Department of Cell Chemistry, Graduate School of Medicine, Dentistry, and Pharmaceutical Sciences, Okayama University, Okayama 700-8558, Japan; p5r62alh@s.okayama-u.ac.jp (M.S.H.L.); hkobuchi@md.okayama-u.ac.jp (H.K.); 2Department of Interdisciplinary Science and Engineering in Health Systems, Okayama University, Okayama 700-8530, Japan; ohtsuk@okayama-u.ac.jp; 3Collaborative Research Centre for OMIC, Graduate School of Medicine, Dentistry, and Pharmaceutical Sciences, Okayama University, Okayama 700-8558, Japan; f_takenaka@outlook.jp (F.T.); kazukoba@okayama-u.ac.jp (K.K.); akehi@cc.okayama-u.ac.jp (M.A.); t-sasaki@cc.okayama-u.ac.jp (T.S.); 4Department of Material Chemistry, Graduate School of Engineering, Kyoto University, Kyoto 615-8510, Japan; uji.hirotaka.3w@kyoto-u.ac.jp; 5Technology Research Laboratory, Shimadzu Corporation, Kyoto 604-8511, Japan; zeki@shimadzu.co.jp; 6Neutron Therapy Research Centre, Okayama University, Okayama 700-8558, Japan

**Keywords:** theranostics, single chain variable fragment of IgG (scFv), drug delivery system (DDS), photodynamic therapy (PDT), PET imaging, accelerated blood clearance (ABC), cell penetrating peptide (CPP), siRNA, ATP-binding cassette subfamily G member 2 (ABCG2)

## Abstract

“Theranostics,” a new concept of medical advances featuring a fusion of therapeutic and diagnostic systems, provides promising prospects in personalized medicine, especially cancer. The theranostics system comprises a novel ^89^Zr-labeled drug delivery system (DDS), derived from the novel biodegradable polymeric micelle, “Lactosome” nanoparticles conjugated with specific shortened IgG variant, and aims to successfully deliver therapeutically effective molecules, such as the apoptosis-inducing small interfering RNA (siRNA) intracellularly while offering simultaneous tumor visualization via PET imaging. A 27 kDa-human single chain variable fragment (scFv) of IgG to establish clinically applicable PET imaging and theranostics in cancer medicine was fabricated to target mesothelin (MSLN), a 40 kDa-differentiation-related cell surface glycoprotein antigen, which is frequently and highly expressed by malignant tumors. This system coupled with the cell penetrating peptide (CPP)-modified and photosensitizer (e.g., 5, 10, 15, 20-tetrakis (4-aminophenyl) porphyrin (TPP))-loaded Lactosome particles for photochemical internalized (PCI) driven intracellular siRNA delivery and the combination of 5-aminolevulinic acid (ALA) photodynamic therapy (PDT) offers a promising nano-theranostic-based cancer therapy via its targeted apoptosis-inducing feature. This review focuses on the combined advances in nanotechnology and material sciences utilizing the “^89^Zr-labeled CPP and TPP-loaded Lactosome particles” and future directions based on important milestones and recent developments in this platform.

## 1. Introduction

Theranostics is a novel term invented for drugs and mechanisms that are used for simultaneous diagnosis and treatment [1] or a simpler definition of diagnosis plus therapy [2]. The first clinical application of “theranostics” was conceptualized in 1946 [3], utilizing the radioactive iodine therapy, [^131^I ] for patients with metastasized thyroid cancer. Thereafter, various radioligand-based therapies including radio-attached monoclonal antibody, small-molecule inhibitors of prostate-specific membrane antigen (PSMA), and α- and β-emitting radioisotopes have gained popularity in early phase clinical trials, particularly in PSMA expressed prostate cancer [4,5,6,7,8]. Cancer is among the top five major diseases that caused millions of deaths in the 20th century and yet remains a challenging disease to treat causing significant morbidity and/or mortality with over 10 million new cases annually [1]. Cancer therapy, for many decades, has relied on the conventional radiotherapy and chemotherapy, which have significant drawbacks and side effects where non-cancerous cells are also greatly affected by chemotherapeutic action [9]. Although recent medical advancements in the form of targeted treatments, early detection, and behavioral changes have improved cancer prognoses, many treatment options are still reported to be ineffective at preventing recurrences. Moreover, the invasive nature, drug resistance, and systemic toxicity side effects of these treatment options are highly disputed [10]. Previous literatures reported that as much as 70% of ovarian cancers and several types of pancreatic cancers have already metastasized even before diagnosis, thus imploring the need for an earlier and more precise method of diagnosis coupled with targeted treatment [11,12]. The state-of-the-art theranostics concept shifts from the conventional one-size-fits-all medicine approach to a more holistic personalized medicine approach. The goal of this therapy is to offer the right treatment, for the right patient, at the right time while providing the right dose with a more targeted and efficient pharmacotherapy profile. When developing these theranostic-based technologies for clinical translation, it is imperative to focus on adequate blood plasma circulation time, specific delivery to cancerous tissues only while successfully evading normal tissues and organs accumulations, lack of an immune response, and simultaneous treatment coupled with non-invasive monitoring for successful drug delivery. Moreover, the delivery system should also be preferably non-invasive, non-toxic, and biodegradable [13].

The principle behind theranostics is the exploitation of specific biological pathways in the human body for diagnostic imaging. The translation of these images is vital in increasing the probability of the targeted therapeutic dose of radiation accumulating at the disease site while limiting damage to the surrounding healthy tissues. Biomolecular imaging is a technique used for imaging biological phenomena in the space or temporal dimension at both cellular and molecular levels. This diagnosis method may be achieved through the fusion of both biological image engineering with molecular and cell biology technology. Examples of such modalities for *in vivo* imaging include positron emission tomography (PET), single photon emission computed tomography (SPECT), nuclear magnetic resonance imaging (MRI), X-ray CT, fluorescence and chemiluminescence imaging equipment, and cancer (malignant tumor) diagnosis and treatment site where all these modalities are utilized separately [13,14,15,16]. Among these diagnosis procedures, both PET and CT can simultaneously capture the position and properties of the cancer lesion by merging the morphological image and the functional image [17]. As far as PET probes for targeting cancer lesions are concerned, ^18^F-fluorodeoxyglucose (^18^F-FDG) is commonly used. It was first synthesized by Brookhaven‘s chemists in 1976 and is still the most widely used radiotracer for brain imaging and cancer diagnosis worldwide. In ^18^F-FDG PET, a phenomenon such as the detection of the inflammatory site besides cancer lesions for organs such as the brain, heart, and liver may occur because the sugar metabolism will be depicted. However, since the probes and their metabolites are excreted by the kidney, there are numerous cases where imaging diagnosis of cancer lesions was found instead in the urinary tract and its vicinity is proven difficult. Although PET probes are usually derived from methionine or thymidine derivatives, which are sometimes used for monitoring amino acid and nucleic acid metabolism, these probes also suffer the same fate as ^18^F-FDG. Therefore, higher specificity PET probes for diagnosing cancer lesions have been heavily researched [7,17,18,19].

Biomolecular imaging is widely used as a guidance for target-specific therapy through tumor location. It is often coupled with monoclonal antibodies to reduce the systemic toxicity while providing synergistic specific antigen targeting effect [9]. Antibodies are important biomolecules pertaining to biological defense. Over the last decade, antibody drugs have grown rapidly as the leading role of “biopharmaceuticals” especially in the oncological field and autoimmune diseases field. Such FDA-approved therapeutic biopharmaceuticals, derived from chimeric, humanized, or murine antibodies are indicated for cancer, transplantation rejection, psoriatic arthritis, rheumatoid arthritis, and Crohn’s disease [20]. However, these immunotherapy-based treatments have demonstrated more positive results against circulating cancer cells compared to solid tumors, as reported by the FDA approved intact antibodies, Rituxan for the treatment of non-Hodgkin lymphoma and Campath and Mylotarg for the treatment of leukemia [21]. Recently, monoclonal antibodies (mAbs) with high affinity and specificity for various disease-related molecules can be readily prepared. The mAbs in single-fragment variable (scFv) entity are superior to conventional antibodies, due to their smaller size and therefore less possibility of generating anti-mouse antibody responses. Furthermore, with the aim of improving both safety and efficacy, techniques for preparing various low molecular IgG variants (such as scFv) have been pursued. The scFv composed only of so-called “variable parts” consisting of both antigen specific complementarity determining regions (CDRs) and framework sequences has been established [22]. Apart from the role as medical therapeutics, another important application of scFv is in diagnostic applications. The custom-designed recombinant scFv provides potential alternatives to these “conventional” immunodiagnostic reagents, which will be further discussed in this review [23]. Therefore, PET imaging utilizing antibody probes has gained numerous attentions as a method to investigate the expression of biomarkers in the field of personalized medicine, especially cancer. 

The nanomedicine-based theranostic approach is one of the most promising strategies to provide efficient and targeted therapies, especially for recalcitrant cancers. To date, numerous classes of nanomaterials have been explored and studied for their potential application in cancer theranostics, including quantum dots, gold-based nanostructures, near-infrared fluorescence (NIRF) dyes, graphene and its derivatives, liposomes, micelles, and magnetic nanoparticles [1,9]. Lately, peptide-based nanoconjugates have gained popularity in therapeutic delivery, not only for cancer [24], but even for the treatment of systemic lupus erythematosus (SLE) [25]. Recently, Kyoto University and Shimadzu Corporation have established and developed a novel amphiphilic micelle type drug delivery system (DDS) [26], consisting of a biodegradable polymer, namely, “Lactosome”, which is less liable to generate an immune response even when administered continuously in vivo [27,28,29]. Lactosome particles are composed of the AB-type (polymer of 1 poly(sarcosine) (hydrophilic) and 1 polylactic acid (hydrophobic)), or of the A_3_B-type (polymer of 3 poly(sarcosine) and 1 polylactic acid). Lactosome has reported superior biodegradability and biocompatibility over the conventional polyethylene glycol (PEG) liposomes [26]. Lactosome particles with an average diameter of 35 nm are DDS carriers with promising prospects for solid tumor accumulation via the enhanced permeability and retention (EPR) effect [27,29,30]. The AB-type Lactosome polymers are suitable for chemically introducing various functional molecules such as IgG variant(s) and chelator(s), while A_3_B-type polymers are suitable for forming micellar type polymeric molecular assemblies with negligible antigenicity. Micellar type DDS carriers of various forms can be prepared by varying components ratio, heat treatment conditions, and association methods of AB-type and A_3_B-type amphiphiles modified with these functional molecules by using the optimal composition or polymer length at the initial stage of Lactosome DDS carrier development. Thus, the applicable DDS functions of the obtained micelle may be further diversified for diagnostic, therapeutic, or drug monitoring purposes. The site and size modifiable capabilities of Lactosome provide a promising pathway to realize the advanced target medical care “theranostics” capable of fusing a safe and biodegradable DDS device while simultaneously conducting PET diagnosis and drug delivery treatment. Besides its abilities to form stable conjugates with antibody variants and chelators, Lactosome DDS carriers have various physical properties to hold and transport both hydrophilic and hydrophobic effector molecules to the lesion. Such effector candidate molecules are composed of several anticancer drugs such as peptides, genes, and also small interfering RNA (siRNA).

One of the prerequisites for an efficient drug delivery system is the ability to be internalized by cells for intracellular action. Although Lactosome particles have demonstrated superior EPR effects *in vivo*, it is not naturally internalized by the cells. To establish the role of Lactosome as a potential DDS carrier, our team has previously modified the A_3_B-type Lactosome particles with an amphiphilic EB1 type cell-penetrating peptide (CPP) and photosensitizer 5,10,15,20-tetraphenyl-21*H*,23*H*-porphyrin (TPP) to improve the in vitro cellular uptake and photoinduced killing ability in mammalian cancer cell lines [31]. CPPs are short peptides, generally rich in cationic residues, which have the ability to internalize various effector molecules such as nanoparticles, proteins, and nucleic acids [32,33,34]. In addition, the photosensitizer TPP [35] was delivered to mammalian cells (including cancer cells) via the CPP-modified Lactosome particles. This cell treatment of photosensitizer and light is called photodynamic therapy (PDT) and is frequently used for killing malignant cancer cells by apoptosis. PDT is a photochemical process for inducing localized tissue apoptosis through the activation of a photosensitizing drug in the target tissue with a light of specific wavelength to the absorption peak of the particular sensitizer in the presence of molecular oxygen to generate reactive oxygen species (ROS), including singlet oxygen (^1^O_2_) [36]. Moreover, photosensitizers have an intrinsic affinity for tumor tissues compared to normal tissues and are therefore clinically used in PDT of various types of cancers [37]. Hence, PDT is considered to be therapeutically non-invasive and safe, which is a promising candidate for clinical cancer treatment, based on improved therapeutic potency through combined chemotherapy-based nanomedicine and PDT [38]. Another promising strategy involving the combination of PDT and gene therapy via the photochemical internalization (PCI) pathway and/or photoinduced endosomal release has been recently pursued [39,40].

The objective of this review is to discuss some major advances in the field of cancer nano-theranostics as shown in Figure 1 by briefly summarizing the development and clinical potential of various Lactosome-based particles modalities while delineating the challenges required to overcome for successful future clinical development and implementation of such cancer theranostics. Each section here emphasizes the safe and non-invasive precision-based diagnosis and treatment focusing on patient wellbeing. 

## 2. ^89^Zr-labeled PET Imaging and Anti-Mesothelin (MSLN) Single Chain Variable Fragment (scFv)

The first major breakthrough in theranostic radiopharmaceutical in the history of nuclear medicine was radioiodine, ^131^I, which was used for the imaging plus therapy in thyroid diseases [3]. Since then, therapeutic nuclear medicine has gained the reputation as the “gold standard” of theranostics [9]. Nuclear targeted therapies play an important role, particularly in patients with advance neuroendocrine tumors, such as neuroblastoma, pheochromocytoma, gastroentero-pancreatic (GEP) tumors, and broncho-pulmonary neuroendocrine tumors. Further, previous studies have reported positive results with radio-ligand therapies in metastatic melanoma and metastatic prostate cancer [41]. The theranostic approach in nuclear medicine is a fusion of both diagnostic imaging and therapy using the same molecule or at least very similar molecules, which are either radio labeled differently or given in different dosages. For instance, γ and βemitting iodine-131(^131^I) and lutetium-177 (^177^Lu) can be used for both imaging and therapy purposes. Different isotopes of the same element, such as iodine-123 (^123^I, gamma emitter) and iodine-131 (^131^I, gamma and beta emitter), are also used for theranostic purposes. Recently, radionuclides such as zirconium-89 (^89^Zr), yttrium-86/ 90 (^86^Y/^90^Y) or terbium isotopes (Tb): ^152^Tb (β^+^ emitter), ^155^Tb (γ emitter), ^149^Tb (α emitter), and ^161^Tb (β^-^ emitter) are gaining popularity in the field of nuclear medicine [19,42,43]. The detection of potential targets is key in predicting whether a patient will benefit from a particular treatment. Theranostics can potentially be helpful in predicting the potential response and eventual toxicity through precise visualization of drug distribution throughout the whole body. However, the safety profile of high cumulative doses of radioactive agents after multiple administration requires practical considerations.

Recent advances in radionuclide therapy, used interdependently with PET and SPECT molecular imaging, are leading a major revolution in the highly complex nature of cancer metastases and a highly specialized and safe modality is warranted to determine the best therapeutic algorithm for each individual patient. To achieve this, potentially safe radionuclides with adequate half-lives are required for real-time pharmacokinetic observations. PET is one of the most sensitive and quantitative platforms in nuclear medicine and is commonly used with ^18^F-FDG. Pet imaging including ^18^F-FDG is indispensable in quantifying the stage of malignancy, therapeutic effects as well as potential therapeutic outputs in the early diagnosis of lesions. However, potential radiotracer of theranostic qualities should be able to be retained in the body long enough for drug delivery and treatment monitoring purposes. This finding would be highly useful for the early determination of therapeutic effects and subsequently therapeutic selection for personalized medicine in patients. The half-life of radionuclide is the time it takes for half the atoms to decay into its daughter nuclide form. The radionuclides commonly used for PET are carbon-11 (^11^C), oxygen-15 (^15^O), and fluorine-18 (^18^F) with a half-life of 20.3 min, 122 sec, and 109.8 min, respectively. These radionuclides have excellent labeling affinity with good quality PET imaging profiles. Moreover, the small amount of radiotracer with short half-lives means that the patient receives a very low radiation dose. However, the systemically administered nanoparticles and antibodies usually remain in the body for tens of hours, or even days. Therefore, these ultra-short half-life radionuclides PET will not be able to grasp the complete distribution and elimination profile of applied drug therapies, especially antibody-based therapeutics. A positron emitter must first fulfill several requirements to be a suitable candidate for immune-PET. Its physical half-life must be compatible with the time required for a mAb to achieve optimal tumor-to-nontumor ratios. For most intact mAbs, this time generally lasts from 2–4 days [44].

Alternatively, the long-lived positron emitter (^89^Zr) has a half-life of 78.4 hr and is an attractive candidate for theranostic radiotracers involving immuno-PET [44]. Compared to ^18^F-FDG, ^89^Zr utilized immuno-PET displays superior specificity because the conventional ^18^F-FDG not only shows increased uptake in tumors but also in normal tissues expressing high metabolic activity. Besides its diagnostic capabilities, PET can also be used to quantify molecular interactions, which is highly attractive when immune-PET is a prerequisite to antibody-based therapy. It is a step-by-step guide to clinicians in selecting which patient may benefit from the mAb treatment before initiating an individualized therapeutic approach. The immune-PET enables the confirmation of tumor targeting and the quantification of mAb accumulation in tumors and tissues [45,46]. Other potential candidates include copper-64 (^64^Cu) radioisotope with a half-life of 12.7 hr and iodine-124 (^124^I) with a half-life of 100.3 hr. These radionuclides can form stable complexes with mAbs *in vivo* and have been extensively researched [18,19,44]. Among these isotopes, ^89^Zr can be obtained with high yield, high radionuclide purity, and low production costs [47]. Moreover, ^89^Zr demonstrated ideal characteristics for optimal image quality and accurate quantification such as the first clinical study involving PET with ^89^Zr-labeled-chimeric mAb, U36, in patients with head and neck squamous cell carcinoma (HNSCC) [44] and the ^89^Zr -trastuzumab PET guided visualization and quantification of HER2-positive lesions in patients with HER-2 positive metastatic breast cancer [19]. The latter study reported that ^89^Zr has the most favorable half-life, allowing antibody imaging of up to seven days after the injection. Moreover, ^89^Zr showed superiority over radio-halogens such as ^124^I by being residualizing and are therefore retained within the target cell even after internalization and intracellular degradation of the tracer resulting in higher uptake in the tumor when an internalized antibody like trastuzumab is used [48]. Conceptually, the Okayama University in collaboration with Sumitomo Heavy Industries, Ltd. ultimately aim to establish a ^89^Zr-labeled-immuno-PET DDS carrier-theranostic system as shown in Figure 2.

The ability of the novel DDS carrier, Lactosome particles, in forming stable conjugates with radionuclides has been established. Lactosome particles have an ability to accumulate in solid tumors, of which specifics will be further described in this paper. The radionuclide therapy utilizing ^131^I-labeled Lactosome particles (^131^I-Lactosome) was found to be an effective antitumor therapy for mammary cancer cells when combined with the local therapy for the percutaneous ethanol injection therapy (PEIT) [49]. Radiofrequency ablation (RFA) and PEIT are frequently performed in combination with chemotherapies such as epirubicin, cisplatin, and 5-fluorouracil. Inflammation biomarkers are slightly raised after RFA or PEIT treatment [50,51]. Since nanoparticles are known to accumulate in the tumor and inflammation regions via the enhanced permeability and retention (EPR) effect [30,52], the accumulation of nanoparticles in solid tumors was enhanced by the simultaneous treatment of RFA or PEIT. The β-decay radioisotope of ^131^I attached to Lactosome was found to accumulate more effectively in the tumor. There are similar reports regarding the synergistic effect of a nanoparticle-based treatment couple with photoimmunotherapy (PIT), which is due to the claimed “super EPR effect” [53,54]. A separate study by Kurihara et al. investigated the role of indium-111 (^111^In) (^111^In-DOTA-Lactosome) for SPECT imaging and ^90^Y (^90^Y-DOTA-Lactosome) for β-ray irradiation for mammary tumors *in vivo*. This study showed that the synergistic effect of ^90^Y-DOTA-Lactosome and DOXIL^®^ (doxorubicin liposome) demonstrated significant suppressive effects on the tumor growth under PEIT induced inflammatory induction due to the super EPR effect [43]. The same team reported that the ^90^Y carrying Lactosome particles may be a possible candidate as therapeutic agent for the currently incurable meningeal dissemination [55]. These findings confirm that the combination of radionuclide therapy using nanoparticles coupled with conventional chemotherapy is more targeted and effective with improved therapeutic performance, which may be useful for future tumor treatment with less stress to the body. Clinically, the first-in-human involving two patients with clinically critical situation consisting of diffuse red marrow infiltration and resistance to other therapies, showed complete remission to actinium-225 (^225^Ac-PSMA-617) therapy [8]. One of the patients was contraindicated with β-emitters treatment while another patient was resistant to lutetium-177 (^177^Lu-PSMA-617). This study highlights the significant life-saving capabilities of the new era of α-emitting radionuclide-based personalized medicine. 

Macromolecules can easily accumulate in tumor tissues and inflammation regions due to the EPR effect because of the leaky blood vessels surrounding these regions. Typically, nanoparticles, including Lactosome particles, take advantage of this abnormal situation to leak out from these vessels to the tumor site and are able to be retained there for a long period of time due to the undeveloped lymphatic system usually observed in malignancies [30,52]. This phenomenon is tissue selective and the nanoparticles do not accumulate in normal cells. This EPR effect maybe useful for localized tissue radiation therapy as discussed above but is insufficient to deliver drugs which target the internal cavity of cells such as protein, pDNA, siRNA, and miRNA. For this, ligands that allow specific binding to target cells that allows subsequent drug delivery is warranted. For this, mesothelin (MSLN) is a promising candidate for tumor-specific therapy. It is overexpressed in various epithelial cancers such as mesothelioma, ovarian, pancreatic, and lung cancers, normally expressed in normal tissues, and limitedly expressed in pleural, pericardial, and peritoneal mesothelial cells [56,57]. Recent studies have also reported that MSLN is expressed in lung adenocarcinomas, uterine serous carcinoma, gastric cancers, triple-negative breast cancer, acute myeloid leukemia, and cholangiocarcinoma [56,58,59,60,61,62,63,64,65,66,67,68]. MSLN is synthesized from a 71 kDa-precursor protein, which is further cleaved by furin to a 31 kDa-megakaryocyte-potentiating factor (MPF) and a 40 kDa-MSLN protein. MSLN (40 kDa) is attached to the plasma membrane by a glycosylphosphatidylinositol-linked (GPI) anchor-binding protein (Figure 3) and is involved in cell adhesion [56,57,58]. Moreover, studies have reported the various roles of MSLN in cell survival/proliferation, tumor progression, and chemoresistance [69]. MSLN is the driving force in peritoneal implantation and metastasis of tumors through its interaction with CA125 or MUC16, an ovarian cancer antigen [70,71,72]. The overexpression of MSLN promotes cancer cell invasion via matrix metalloproteases 7 and 9 induction and promotes cancer cell survival and proliferation via the NF-*_K_*B pathway [73,74,75]. Overexpression of MSLN activates NF-*_K_*B, which leads to higher interleukin-6 production and subsequently induces tumorigenesis [76]. Furthermore, MSLN expression promotes resistance to certain chemotherapy drugs, such as TNF-α, paclitaxel, and a combination of platinum and cyclophosphamide [77,78]. MSLN also serves as a biomarker of neoplastic transformation of pancreatic epithelial cells and is overly expressed in 86% of pancreatic cancer patients [57]. MSLN has been attracting attention as a cancer differentiation antigen due to its high expression in several malignancies and is currently being evaluated as a target for antibody- and vaccine-based therapies for cancer [58,79]. The MSLN protein is the antigen recognized by the mAb, K1, and is a good target for tumor-specific antibody-based therapies [80]. To date, several anti-MSLN mAbs have been developed, such as MORAb-009 [71], commercially known as Amatuximab, which is a chimeric (mouse/human) antibody, and HN1 [81], which was isolated from a human scFv phage display library and converted into a fully intact human IgG. Our team has also demonstrated the specificity of the ^89^Zr-labeled anti-MSLN-mAb in MSLN-positive and MSLN-negative tumor mice model via PET imaging compared to the ^89^Zr- and ICG-labelled Lactosome particles (Figure 2), spearheading the possibility of ^89^Zr-labeled anti-MSLN-mAb as a novel antibody-based therapy.

Following our aim to develop a specific antibody-based PET probe that can detect a wide range of cancers, our team has examined the role of IgG anti-MSLN (11–25) mAb, which is derived from a murine hybridoma as an imaging probe for detecting MSLN-expression tumors. Our study has demonstrated that the 11–25 mAb specifically detected MSLN by ELISA and by flow cytometry/immunohistochemistry and furthermore *in vivo* NIRF imaging. The ^64^Cu-labeled 11–25 mAb significantly accumulated in MSLN-expressing tumors in PET imaging and biodistribution studies with the preservation of immunoreactivity and thus confirming its potential for future diagnosis of patients with MSLN-positive cancers, such as malignant mesotheliomas, ovarian cancers, and pancreatic cancers [18]. However, a few drawbacks in this study should be addressed: 1. Due to the short half-life of ^64^Cu (12.7 hr), the biodistribution of ^64^Cu-DOTA-11-25 mAb can only be observed for a maximum of 48 hr post-injection, and 2. the relatively high accumulation of ^64^Cu-DOTA-11-25 mAb in the liver, as previously reported with ^64^Cu-labeled IgG [82]. There are two possible reasons for this: 1. The nonspecific binding of IgG to the presented Fc receptor in the liver [83] and 2. the transchelation of ^64^Cu to superoxide dismutase and metallothionein [84]. Therefore, the development of smaller size fragmented antibody derivatives such as Fab and scFv would be useful in achieving faster blood clearance and improved “target-to-non-target” contrast at an early time point after administration compared to that of IgG 11-25 mAb. For these reasons, our research team has demonstrated the utilization of ^89^Zr-labeled anti-MSLN-scFv as a potential rapid PET probe for MSLN-positive cancer diagnosis [85]. In this study, the PET imaging using the ^89^Zr-labeled anti-MSLN-scFv clone (H1a050) with high reactivity to MSLN-expressing cancer cells clearly displayed the MSLN-positive tumor derived from NCI-N87 cancer cells but not the MSLN-negative tumor derived from PANC-1 cells (Figure 4). According to the blood and tumors distribution analysis, the ^89^Zr-labeled anti-MSLN-scFv was rapidly discharged from the blood and retained in tumors even at 3 hr post-injection. Hence, this study concluded that the newly discovered anti-MSLN-scFv enabled clear visualization of MSLN-expressed tumors via PET imaging at just 3 hr post-injection, suggesting that it may be a useful candidate for tumor imaging [85].

## 3. Polymeric Micelle-Type DDS Carrier as the “Core” of Theranostics Technology

Theranostic nanomedicine for medical purposes consists of colloidal nanoparticles ranging in sizes from 10 to 1,000 nm (1 µm). They include macromolecular materials/polymers in which the diagnostic and therapeutic agents are adsorbed, conjugated, entrapped, and encapsulated for diagnosis and treatment, simultaneously at both cellular and molecular level [86]. Theranostic nanomedicine is superior compared to conventional theranostics because they have advanced capabilities as an all-in-one single platform, which include targeted delivery, sustained/controlled release, higher transport efficiency via endocytosis, remotely triggered delivery, synergistic performance (e.g., siRNA co-delivery, PDT and PCI co-delivery, and chemotherapy combination therapy), and multimodality diagnosis and/or therapies and quality performances (e.g., oral delivery, evasion of multi-drug resistance (MDR) protein, and autophagy inhibition) [9]. There are several prerequisites for a successful nano-theranostics agent: 1. The nanocarrier should be easily manufactured using standard procedures in nanotechnology to provide designed functionalities and achieve specific targeting. 2. Nanomolecules must be efficient enough to improve the pharmacokinetic profile while enhancing the biodistribution of existing therapeutic moieties to the targeted site. 3. They must have promising intrinsic advantages for site specific delivery within solid tumors via the leaky vasculature [87]. 4. Nanocarriers should be non-toxic and biocompatible polymers to improve the safety index of the anticancer agent to reduce overall systemic toxicity. 5. The nano system should possess the inherent advantage of enhancing aqueous solubility of lipophilic compounds so that they may be appropriate for parenteral administration. 6. They should have a good stability profile when loading or encapsulating therapeutic entities such as small-sized hydrophobic molecules, peptide drugs, and even oligonucleotides [88]. 7. The nanomolecules should demonstrate little uptake into the reticuloendothelial system (RES) such as the liver and spleen. 8. The nanoparticle should also be stealthy enough to evade recognition by the immune system [89].

Among these prerequisites, stealth to evade recognition by the immune system is a major challenge for emerging nanoparticles. The first approved “PEGylated” products have been in the market for over 30 years still undergo severe interaction with the immune system when applied systemically. It was reported that the subcutaneous injection of an PEG-modified ragweed allergen in humans triggered the formation of IgM isotype antibodies to PEG and pre-existing IgG and IgM anti-PEG antibodies were identified in more than 25% of healthy donors [90,91]. The presence of these anti-PEG antibodies was strongly correlated to the rapid blood clearance of PEG conjugates, which inadvertently affects the pharmacokinetic profile of these conjugates [92]. Although considered as the gold standard in the field of polymeric drug delivery, PEG displays a vast amount of other side effects and complications such as its nonbiodegradability, unexpected changes in pharmacokinetic behavior, toxic by-products, and an antagonism arising from the easy degradation of the polymer under mechanical stress [89]. There were even reports on anaphylactic shock induced by PEG [93,94,95]. Since then, a variety of promising hydrophilic polymers have emerged as synthetic alternatives to PEG. Among them is the novel amphiphilic block polymers composed of poly(**_L_**-lactic acid) (PLLA) and poly(sarcosine), molecular assemblies with various morphologies such as micelle, vesicle, and lamella were formed and collectively named as “Lactosome” particles [26].

The amphiphilic poly(sarcosine)-*b*-poly(**_L_**-lactic acid) (Lactosome) is an established polymeric micelle with an average diameter of ca. 35 nm [96,97]. Sarcosine (Sar) or N-methyl glycine is a natural amino acid and its homopolymer shows high solubility against aqueous solution like PEG [98]. Hence, polymeric micelles with their surfaces covered with poly(sarcosine) are expected to show prolonged blood circulation characteristics with negligible undesired accumulation by the RES [99]. The poly(**_L_**-lactic acid) or PLLA component, on the other hand, is one of the most commonly used biodegradable and bioinert materials [100,101] and is known to form 3_10_ helical structure [102]. In the concept of radionuclide for PET imaging, the [^18^F]SFB labeled poly(**_L_**-lactic acid) of 30mer, consisting of the hydrophobized ^18^F attached to the PLLA chain, was encapsulated into the core/hydrophobic region of Lactosome via hydrophobic interactions [103]. This ^18^F labeled Lactosome particles, which was administered *in vivo* via the tail vein of the tumor transplanted mice, displayed clear PET images post 6 h of dosage. Here, Lactosome particles showed excellent blood circulation aspects owing to the surface modification with hydrophilic poly(sarcosine) chains. Hence, the signal intensity at the organs with high blood flows was high and the accumulated signal in the transplanted tumor could be detected. Since the surface of the Lactosome particles was not specifically modified by ligands in this study, Lactosome particles were considered to be passively accumulated to the tumor region by the EPR effect [103]. In the concept of therapeutic radionuclide, studies have been conducted on the *N*-succinimidyl 3-[^131^I]iodobenzoate ([^131^I]SIB) labeled PLLA encapsulated into the Lactosome particles as β-ray emitter by the same approach with the previous ^18^F compound for PET imaging [49]. To the preliminary PEIT treated tumor transplanted mice, ^131^I labeled Lactosome particles of 200 MBq/kg were injected and the time course of tumor growth was investigated. It was concluded in this study that the ^131^I labeled Lactosome particles provided efficient delivery to the tumor region with significant tumor suppression [49]. Lactosome demonstrated excellent EPR effects in numerous cancer cells such as human pancreatic cancer, hepatocellular carcinoma, lung papillary adenocarcinoma, mouse 4T1 breast tumor, and even rat mammary adenocarcinoma [49,96,103,104]. Besides tumor imaging [96,103] and anti-tumor therapy [49], it has also been applied for tumor-selective PDT [104] which will be further discussed later.

Although Lactosome particles showed promising capabilities as a nanocarrier for drug and/or imaging agent delivery, it is not without any shortcomings. For drug delivery and treatment purposes, nanocarriers are expected to show unaltered disposition upon multiple administrations. However, the production of anti-Lactosome antibody was observed not only three days after the first administration, but it remained elevated for a duration of six months [29]. As for the second dosage of Lactosome particles’ administration, it was immediately opsonized by the anti-Lactosome antibody soon after administration and subsequently entrapped by the RES. This phenomenon is commonly generalized as the accelerated blood clearance (ABC) phenomenon of Lactosome particles, which is comparable to the phenomenon sometimes observed on PEGylated materials [105,106]. Lactosome, together with the PEGylated liposomes, belong to the T cell-independent (TI) type II antigen [29,107], which are antigens that can trigger the antibody production without the help of T cells [108,109]. Previous research has reported that the peritoneal B1a cells recognize a TI type II antigen of Lactosome and hence responsible for the ABC phenomenon [110]. On the other hand, this interaction was not observed in PEGylated liposomes where the splenic marginal zone B (MZ-B) cells [111] are responsible for anti-PEG IgM production for PEGylated liposomes [112,113]. As the antibody-secreting cells (ASCs) are found only in the spleen and bone marrow, but not in the peritoneal cavity (PerC), there is therefore a migration of Lactosome-responsive B1a cells from the PerC to the spleen upon Lactosome stimulation [110]. This difference in the trigger of B1 subsets may be related to the long-lasting memory and IgG_3_ production induced by Lactosome but not by PEGylated liposomes. These reports indicate that the ABC phenomenon for different nanomolecules is driven by different immune system pathways, which should be further investigated to improve the efficacy and safety profile of potential nanotherapeutics.

Up to date, there have been extensive studies to overcome this problem. Similar to TI antigens such as PEGylated liposome, which activate B cells and induce IgM antibodies production at the early stage of administration, however, at high doses, lack the activation ability on B cells [114,115]. This phenomenon is called immune tolerance and is widely used in PEG derived therapeutic where the ABC phenomenon is suppressed by high-dose PEGylated liposomes [105,116,117]. In one study where a wide range of Lactosome dose (0–350 mg/kg) was administered in tumor transplanted mice, the production amounts of anti-Lactosome IgM were found to be inversely correlated with that of the first Lactosome dosage. The first Lactosome dose dependence of the ABC phenomenon was reported at a critical value of 50 mg/kg dose [118]. If the first administered Lactosome dose is below 50 mg/kg, the Lactosome at the second administration will be entrapped in the liver. However, when a high first Lactosome dosage was administered (>150 mg/kg), the Lactosome was found to be accumulated in the tumor region by the EPR effect indicating that the Lactosome ABC phenomenon was suppressed by the induced immunological tolerance effect [118]. However, the anti-Lactosome IgM in serum remained high irrespective of second doses indicated that the ABC phenomenon still occurred, but the anti-Lactosome IgM was saturated with excessive doses of over 50 mg/kg at the second Lactosome administration. These lead to ICG-Lactosome free from binding of anti-Lactosome IgM and to efficiently accumulate in the tumor. Overall, we interpreted that with the increment of first Lactosome doses, the anti-Lactosome IgM level decreases gradually due to partial immune tolerance. When this anti-Lactosome IgM level becomes lower than the amount required for opsonization of Lactosome dose as low as the amount of the second dose, the ABC phenomenon disappears upon the second administration of the Lactosome particles. More importantly, there was no acute toxicity induced even at such high Lactosome dosage administration [118]. Therefore, a high dose Lactosome administration approach may be adopted to evade the Lactosome ABC phenomenon, which makes it possible for Lactosome to be used for multiple imaging.

Besides using high doses of Lactosome to induce immunological tolerance, further studies were conducted on the suppression of anti-Lactosome IgM by modification of the lactosome structure itself. Several studies have reported on the improved stealth effect of Lactosome from the immune system when the nanoparticle surface is covered densely by hydrophilic polymer chains such as a high-density polymer brush structure. This dense hydrophilic shell may evade capture by B-cell receptors, thus preventing B-cell receptor recognition [27,28,119]. Spherical structures of micelles and small vesicles have a disadvantage because of the existence of their relatively looser spaces at the outer surface compared to the inner regions due to the large curvature of the molecule [27]. Hence, a sheet structure of ~20 nm was prepared from poly(Sar)_m_-*block*-(**_L_**-Leu-Aib)_n_, which consists of a series of amphiphilic helical peptides and a hydrophobic helical block of (Leu-Aib)_n_ (Aib represents α-aminoisobutyric acid) [120]. This peptide nanosheet, despite having the same poly(sarcosine) chains as Lactosome, did not induce the ABC phenomenon where it completely loses any epitope activity due to the high-density polymer brush state [27]. Therefore, the formation of a high-density polymer brush on the surface of nanoparticles is one of the many strategies for nanoparticles to elude the immune system [26]. 

It was reported that when the surface modifying polymer chains become a high-density polymer brush state corresponding to a surface density of 0.3 chains/nm^2^ or over, the surface provides a lubricating surface which also serves as an antifouling property [121,122]. Thus, a novel amphiphilic polydepsipeptides with three hydrophilic branch chains connected to one hydrophobic poly(lactic acid) chain (A_3_B-type) was developed as shown in Figure 5 [28]. The A_3_B-type polymeric micelle size went down to ca. 22 nm from the AB-type of ca. 36 nm [123]. Notably, the A_3_B-type Lactosome attenuated the ABC phenomenon compared to AB-type Lactosome as observed by the NIRF imaging and anti-Lactosome IgM production (Figure 6) [28]. The anti-lactosome IgM production was significantly lower with A_3_B-type Lactosome compared to AB-type Lactosome, indicating lower immunogenicity profile of the A_3_B-type Lactosome. Figure 6B shows that the AB-type Lactosome, upon first administration, spread over the whole body and gradually accumulated in the tumor region via the EPR effect. However, upon second administration (after seven days) the AB-type Lactosome immediately accumulated in the liver as previously reported (Figure 6D) [29]. On the other hand, A_3_B-type Lactosome was observed to spread over the whole body to accumulate in the tumor region upon both first and second administration as shown in Figure 6C, E. There are several ABC attenuation factors regarding this: 1. By increasing the local density of the poly(sarcosine) chains on the micelle surface, which is 0.30 chain/nm^2^ for the A_3_B-type Lactosome, which is 4 times higher than 0.07 chain/nm^2^ for the AB-type Lactosome, the higher surrounding hydrophilic polymer chains successfully prevent the interaction between the micelles and B cell receptors [28]. 2. The half-life of A_3_B-type Lactosome in the bloodstream was 4.3 hr, which is considerably shorter compared to the AB-type Lactosome of 17.2 hr. This may result in the nanoparticle having less chance to trigger the immune system. Unfortunately, a shorter lifetime in the bloodstream is also reflected in a decrease in tumor accumulation effect by the A_3_B-type Lactosome particles. Therefore, the optimal chain length of poly(sarcosine) and composition of lactosome to stabilize the polymeric micelle structure were investigated [119]. Kurihara et al. reported that the optimized poly(sarcosine) chain length lies between 33mer and 55mer for ABC phenomenon suppression. There are several explanations for this: 1. These modified A_3_B-type Lactosome particles showed relatively longer blood half-life, between 6.0 and 9.2 hr compared to the shorter 23mer A_3_B-type Lactosome of 4.3 hr [28]. 2. After 48 hr of administration, Lactosome with longer poly(sarcosine) chain lengths above 33mer presented less than 20% dose of liver accumulation compared to 40–50% dose for Lactosome with 10mer poly(sarcosine) chain length [119]. 3. The tumor/liver accumulation ratio was double compared to Lactosome with 10mer poly(sarcosine) chain length. 4. The antigenicity of 33mer poly(sarcosine) chain length Lactosome was reduced at least 3 times compared to the AB-type Lactosome upon multiple administration (40% of anti-Lactosome IgM produced by AB-type Lactosome at first administration and about 35 to 50% of that by AB-type Lactosome on second administration). From these findings, we can confirm that it is the A_3_B Lactosome structure as a whole rather than the surface density of the hydrophilic polymer chain that determines its stealth index. Hence, the optimum poly(sarcosine) chain length between 33mer and 55mer to lower the antigenicity of the nanoparticle while maintaining the *in vivo* disposition was elucidated.

## 4. Lactosome Mediated RNAi.

It has been established that Lactosome has various advantages over PEG in terms of biodegradability, their ability to escape from the RES, their EPR effect, and their relatively long half-life in the bloodstream. Moreover, the size-controllable and easy modifiable characteristics of Lactosome makes it a powerful tool in both targeting and imaging applications as previously discussed. However, although Lactosome particles can selectively accumulate around solid tumors, it is not efficiently internalized by cells [31]. Therefore, improving the cellular uptake of Lactosome particles is key for efficient delivery of drugs that acts intracellularly, such as siRNA. RNA interference (RNAi) was discovered over two decades ago and showed promising prospects in treating genetic diseases, including cancer, by means of altered gene expression [124,125,126]. SiRNA, in particular, has several therapeutic advantages over conventional chemotherapeutic anti-cancer agents. One advantage is its high specificity with minimal toxicity through siRNA-mediated gene silencing in overexpressed genes presented in recalcitrant cancer progression and metastasis [127,128]. Moreover, the versatility of siRNA due to its unlimited choice of complementary base pairing targets [129] and sustainability of the RNA-induced silencing complex (RISC) for messenger RNA (mRNA) degradation [130] makes it a powerful tool in therapeutic application, especially in the context of personalized medicine.

However, the extensive use of siRNA clinically is encumbered by several limitations at the cellular level: Inability to readily cross biological membranes due to its intrinsic anionic charge, poor endosomal escape leading to cytosolic delivery failure, and lack of target specificity [130,131,132]. Additionally, siRNA is physiologically unstable and susceptible to degradation by serum nucleases, renal elimination, and mononuclear phagocytic system uptake when delivered systemically. Therefore, the encapsulation of siRNA into vesicles is vital for efficient cell and systemic delivery. The successful application of siRNA for cancer therapy mandates the development of both biodegradable and sustainable DDS as the unique physiological aspects of solid tumors render numerous challenges for successful therapy [38,130,131,133]. To accomplish this, our team has modified the A_3_B-type Lactosome particles with CPPs to increase the cellular uptake efficiencies of Lactosome [31]. Cationic CPPs have been widely used for the intracellular delivery of macromolecules including proteins (e.g., antibodies), plasmid DNA, peptides, and antisense oligonucleotides [134]. In addition to the conventional endocytosis pathway, CPP-mediated siRNA delivery systems may penetrate cells directly by crossing the cell membrane [124]. The plasma membrane is a selectively permeable barrier of living cells, which is vital for cell survival and function. In many cases, the efficient passage of exogenous bioactive molecules though the plasma membrane still remains a major predicament for intracellular delivery of cargo. Hence, molecular transporters have long been pursued that would enhance the transportation efficiency of therapeutic and imaging agents into living cells [32]. CPPs are a class of diverse peptides, typically with 5–30 amino acids, that are among one of the most popular and efficient vectors for achieving cellular uptake via direct crossing of the cellular membrane [135,136]. The first CPP was discovered in 1988 and the concept of protein transduction into cell was presented by Frankel and Green. This led to the discovery of the transactivator of transcription (TAT) protein of HIV, which can cross the cell membranes to be efficiently internalized by cells in vitro, resulting in the transactivation of the viral promoter [137]. Since then, various CPPs have been developed for a variety of applications. CPPs serving as vectors can successfully deliver intracellular cargoes such as siRNA [138,139], nucleic acids [140,141], small molecule therapeutic agents [142,143], proteins [144], quantum dots [145], and MRI contrast agents [146], both in vitro and in vivo. Moreover, CPPs are relatively safe and efficient transport system with lower toxicity compared to other delivery methods [147]. CPPs not only facilitates the cellular uptake of various cargoes without any cellular injury, they can also be taken up by cells via multiple pathways, such as direct translocation through the membrane bilayer and endocytosis-mediated uptake. Therefore, depending on the context of experimental conditions, CPP potential for variable modification design can be exploited to harness alternate mechanisms of uptake and sometimes these different pathways may even operate concurrently [147]. Therefore, CPPs have received tremendous attention for medical applications.

A previous study has reported that the CPP-modified A_3_B-type Lactosome particles showed significant improvement in cellular uptake. Several CPPs were investigated including Tat, PTD4, DPV3, MPG^ΔNLS^, R9MPG, Pep1, and EB1. Among these, amphipathic EB1 and Pep1 peptides showed marked improvement on the uptake efficiency of Lactosome particles as shown in Figure 7 [31]. The CPPs were conjugated to the A_3_B-type Lactosome via maleimide thiol reaction (Figure 7). Results show that the CPP-modified Lactosome particles internalized by cells were localized mainly in the endosome and lysosomes. Hence, PDT experiments were conducted using CPP-modified and photosensitizer (TPP)-loaded Lactosome particles. Cell killing was efficiently photoinduced using EB1/TPP and Pep1/TPP Lactosome particles confirming the role of the CPP-modified lactosome as a DDS. However, in vivo imaging of the EB1/TPP/ICG-lactosome showed that this modified Lactosome particles accumulated in NCI-N87 tumors in mice, which is a human gastric cancer, while only slight ICG fluorescence was detected in PANC-1 tumor which is a human pancreatic carcinoma. Pancreatic tumors are known to be poorly permeable, and among polymeric micelles with diameters of 30, 50, 70, and 100 nm, only the 30 nm micelles were accumulated in pancreatic tumors [148]. Since the size of the EB1-modified Lactosome was around 38 nm, more efficient tumor accumulation may be accomplished through the size-control of the CPP-modified Lactosome particles. The CPP-modified Lactosome successfully encapsulate the hydrophobic agent TPP and deliver it into the cells. Hence, with this modification, it is anticipated that such Lactosome particles may also be able to deliver hydrophilic drugs such as proteins and siRNA by attaching hydrophobic modifications to them.

The CPP-modified Lactosome particles capabilities to deliver hydrophilic drugs such as siRNA has also been established. We have successfully conjugated the ATP-binding cassette transporter G2 (ABCG2) siRNA to the L7EB1 type CPP modified Lactosome particles via disulfide bonds to improve RNA stability and transfection efficiency. In this study, the addition of hydrophobic (Leu)_7_ residues to EB1 peptide, collectively known as L7EB1 enables its assembly into Lactosome particles (Figure 8). ABCG2, also formerly known as the “breast cancer resistance protein” [149]*,* belongs to a group of transporters capable of elucidating multidrug resistance (MDR) leading to chemotherapy failure [150]. ABCG2 is ubiquitously expressed in normal tissues while overexpressed in various cancer cells. ABCG2 also plays a role in photosensitivity and phototoxicity regulation through the accumulation of porphyrin [151,152] which will be further elaborated in the next section. ABCG2 expression is therefore inversely correlated with porphyrin derivatives in cancer cell lines and is a precursor for photodynamic diagnosis (PDD) and ALA, which provides an interesting platform for our study as our previous study focuses on the PDT mediated cell killing using EB1/TPP and Pep1/TPP Lactosome particles. After complexing the L7EB1-modified A_3_B Lactosome particles loaded with the photosensitizer 5,10,15,20-tetra-kis(pentafluorophenyl)porphyrin (TPFPP) to the hydrophobically modified ABCG2 siRNA (L7EB1/TPFPP/siRNA-Lactosome particles) [153] and coupled with short photoirradiation, efficient gene silencing was demonstrated via the enhanced cellular uptake and PCI-induced endosomal membrane disruption. Under non-toxic conditions, utilizing the L7EB1/TPFPP/siABCG2-Lactosome particles, our study reported a knockdown efficiency of up to 76.1 ± 8.6% and 84.4 ± 3.7% for stably expressed PANC-1 and NCI-H226 cells, respectively (Figure 8). This observation confirms the synergistic mechanism of the L7EB1/TPFPP/siRNA-Lactosome particles as a whole is mandatory for efficient siRNA silencing in cancer cells. It is obvious that the two therapies operated sequentially: 1. The incorporation of L7EB1 for cellular uptake efficiency and 2. The PCI of TPFPP photosensitizer for enabling photo-induced endosomal release of siRNA. Furthermore, the L7EB1/TPFPP/siRNA-Lactosome particles did not exhibit any photo-induced cytotoxicity on their own, even after continuous light irradiation of 405 nm for 20 s [153]. The PCI strategy has been widely applied to release a spectrum of macromolecules such as toxins and DNA delivered within a complex of cationic polymers or adenovirus or adeno-associated viruses, dendrimer-doxorubicin conjugates, peptide nucleic acids, and even bleomycin, from endosome to cytosol [154,155,156,157]. Thus, this novel CPP-modified Lactosome particles loaded with photosensitizer provides the necessary characteristics for a successful PCI-modulated intracellular siRNA delivery.

## 5. ABCG2 Knockdown by RNAi

ABCG2 was originally cloned from breast tumor cell lines and placental tissue in 1998 [158,159,160] and its isolation from a multidrug resistant breast cancer (Mcf7-derived) cell line led to its latest term as the ‘breast cancer resistance protein.’ MDR is the most frequent phenomenon associated with the failure of chemotherapy. Several cancers have shown to have poorer prognosis if there is either a pre-existing MDR pump, or if the expression of the MDR pump develops because of chemotherapy. ABCG2 protein is one of the several human ATP binding cassette (ABC) proteins capable of MDR [149]. Therefore, for the past decades, the use of nanotechnology to overcome MDR has gained the deliberation of researchers worldwide. Albeit previously isolated from multidrug resistant cancer cells, the expression and distribution pattern of ABCG2 in normal cells and tissues substantiates its other important physiological role, which is protecting the organism as a first line of defense against environmental toxins. Several studies have confirmed this role on ABCG2-null mice where these animals are more susceptible to diet-induced protoporphyria and phototoxicity, caused by the accumulation of pheophorbide, a chlorophyll degradation product, and a confirmed ABCG2 transport substrate [152,161]. Moreover, in both humans and rodents, studies have shown that the localization of ABCG2 carried a vital role in limiting absorption (in the small intestine), mediating distribution (e.g., blood–brain and blood–placental barriers), and facilitating elimination and excretion (liver and kidney) of drugs or xenobiotics that are ABCG2 transport substrates [162,163]. Therefore, more extensive studies to examine the role of ABCG2, especially in solid tumors, is warranted to provide a constructive picture of this pump’s role in cancer biology.

It is widely known that the ABCG2 expressed in the mitochondria is responsible in maintaining low concentrations of anticancer drugs. Moreover, accumulating evidence indicates that ABCG2 plays a particularly important role in regulating the cellular accumulation of porphyrin derivatives in cancer cells, thereby affecting the efficacy of photodynamic diagnosis (PDD) and PDT [151]. PDT for cancer patients is a safe and efficient clinical treatment modality involving the administration of a tumor-localizing photosensitizer or photosensitizer prodrug (5-aminolevulinic acid [ALA], a precursor in the heme biosynthetic pathway) and the subsequent activation of the photosensitizer by light. Among the photosensitizers used clinically, ALA-derived protoporphyrin IX (PpIX) has been the most active area of clinical PDT research [164]. Therefore, the application of 5-aminolevulinic acid (ALA)-based (PDT), where protoporphyrin IX (PpIX) is the main photosensitizer, is a promising strategy for a wide range of cancers. There are several advantages regarding ALA-PDT: 1. Photosensitization lasts for only 24 hr due to the rapid excretion of PpIX and its rapid conversion to heme [165], compared to photofrin, which are retained in the skin tissue for four to six weeks [166]. 2. Following ALA treatment, PpIX is selectively accumulated in the tumor tissues [167]. 3. Topical ALA-PDT is non-invasive and convenient for the treatment of outpatients, having no interaction with other medications, no systemic toxicity, and no cosmetic side effects compared to conventional cancer therapies [164]. However, this treatment method is ineffective in cancers that exhibit poor PpIX accumulation. 

Successful ALA-induced PpIX accumulation heavily depends on the activity of enzymes that synthesize and metabolizes PpIX and on the protein that transport PpIX [168,169]. Exogenously added ALA is taken up by target cells and metabolized to coproporphyrinogen III in the cytosol by several enzymes, including porphobilinogen deaminase, uroporphyrinogen III synthase (the rate-limiting enzyme of porphyrin metabolism), and uroporphyrinogen decarboxylase. Coproporphyrinogen III is then translocated into mitochondria through the ATP-binding cassette transporter B6 and metabolized to PpIX by coproporphyrinogen oxidase and protoporphyrinogen oxidase. PpIX is then metabolized further by ferrochelatase (FECH) to heme (Figure 9). Furthermore, accumulating evidence reported that PpIX appears to be pumped out of cancer cells by ABCG2, which is associated with multidrug resistance-associated protein [169,170,171]. Therefore, heme synthesis enzymes and ABCG2 play vital roles in regulating the cellular accumulation of PpIX in cancer (Figure 9). Previous studies from our team have addressed the mechanism of photosensitizer accumulation to expand PDT applications. A study by Ogino et al. reported that a small amount of serum in medium can substantially decrease PpIX content in T24 cells, which is a human urothelial carcinoma cell line. The albumin content in the serum appeared to be responsible for this effect. As ABCG2 was highly expressed in T24 cells, and it was reported that ABCG2 transported porphyrin derivatives, including PpIX to extracellular albumin through its large extracellular loop ECL3 [172], ABCG2 was deduced to be the principal transporter of PpIX in T24 cells [173]. This study concluded that ABCG2-mediated PpIX efflux was the most important in control cells, and when this export system was abolished, then heme synthesis becomes a primary factor that decrease cellular PpIX. When considering PpIX accumulation driven PDT for malignancies, cell-to-cell variation and subcellular localization of PpIX is important because PpIX negative cells may be immune from PDT. Therefore, to minimize cancer cells that escape PDT, the enhancement of PpIX accumulation is an attractive solution [173].

Subcellular localization of PpIX determines how the cell dies on light illumination and previous reports have suggested that preferential accumulation of PpIX in mitochondria favors apoptosis, whereas PpIX outside mitochondria favors necrosis [174]. Thereafter, our team demonstrated that ABCG2 is predominantly co-localized with mitochondria, hence suggesting that ABCG2 distributed in the mitochondrial fraction plays an important role in the regulatory mechanism of ALA-mediated PpIX accumulation as depicted in Figure 9 [175]. The majority of ABC transporters localize to the ER-derived secretory organelles, including the ER, Golgi apparatus, lysosomes, and plasma membrane. These ABC transporters are initially targeted to the ER by signal sequence, where they are integrated into the ER membrane, sorted, and directed to their final destinations via vesicle transport. ABCG2 in particular has been demonstrated to localize at the plasma membrane to export intracellular heme and porphyrins [176]. Our study demonstrated that ABCG2 was distributed not only in the ER-derived membranes (ER, Golgi apparatus, and plasma membrane), but also in the mitochondria of A549 (human lung adenocarcinoma cell line) and ST-HEK cells stably transfected with FLAG-ABCG2. Moreover, the ABCG2 is expressed in the mitochondrial cristae and is found to be functionally active because the isolated mitochondria these cells accumulated exogenously added doxorubicin and then exported it via ABCG2 [175]. Overall, this study concluded that ALA-mediated PpIX accumulation was found to be negatively correlated with the PpIX transporter ABCG2, but not with that of PEPT1, PEPT2, or FECH.

A separate study demonstrated the use of genistein, a phytoestrogen that appears to competitively inhibit ABCG2 for the enhancement of ALA-induced PpIX accumulation [177]. This study showed that genistein promoted the accumulation of PpIX in A549 cell lines both in vitro and in vivo by preventing ABCG2-mediated PpIX efflux and upregulating the gene expression of heme synthesis enzymes. Estrogen-depleted rats were found to reduce ALA-induced PpIX levels in their tumors suggesting that estrogen and the phytoestrogen genistein may accelerate the accumulation of PpIX via ERβ signaling. This study reported that ALA-induced PpIX accumulation was increased by 3.7 times *in vitro* and 1.8 times *in vivo*, following pre-treatment with genistein [177]. Although PpIX was found to selectively accumulate in tumors [164], ALA-based PDD and PDT remained unsatisfactory in diagnosing some tumors that accumulate insufficient amounts of PpIX. Recently, advancements in nanopreparations to overcome multidrug resistance in cancer have been pursued [129]. For instance, a study involving PDT using a photosensitizer-encapsulated polymeric nanoparticle was reported to be able to overcome ABCG2 induced drug resistance in pancreatic tumors [178]. This study also emphasized the importance of controlling the expression of ABCG2 to attain the intracellular concentrations of a photosensitizer necessary for efficient PDT. Moreover, our recent study demonstrated that the L7EB1/TPFPP/siABCG2-Lactosome particles, not only exhibited an ABCG2 gene silencing effect in the cytosol, but also mediated photo-induced cell death via the ALA-mediated PpIX accumulated PDT pathways in cancer cells [153]. After 48 hr of Lactosome-particles transfection with ABCG2 siRNA, when exogenous ALA was added to the treated cells coupled with PDT treatment, significant reduction in cell viability was observed in both PANC-1 and NCI-H226 cell lines. These observations substantiate the negative correlation of PpIX transporter ABCG2 expression level to ALA-mediated PpIX accumulation, and thereby facilitating the efficacy of PDT therapy in cancer cells [153]. Herein, a safe and efficient siRNA conjugated polyplex was developed for as a synergistic treatment for recalcitrant cancer cells through the integration of PCI-mediated gene silencing and PDT.

## 6. Future Perspective and Limitations

Theranostics is the fusion of multiple advanced elemental technologies. Up to now, our team has established the diagnostic part of theranostics involving the specificity of anti-MSLN antibody and combining it with the sensitivity and high resolution of PET imaging. For the therapeutic part, an enhanced cellular uptake CPP-modified and photosensitizer (TPP)-loaded Lactosome particles was developed to deliver and induce RNAi via light irradiation was developed. Moreover, we have also demonstrated and confirmed that ALA-mediated PpIX accumulation was negatively correlated with ABCG2 expression level, suggesting that the knock down of ABCG2 may increase the accumulation of PpIX in cancer cells and thus selectively induces apoptosis. Subsequently, the ABCG2 knockdown via RNAi was demonstrated using the L7EB1/TPFPP/siABCG2-Lactosome particles, which exhibited both ABCG2 gene silencing with synergistic photo-induced cell death via the ALA-mediated PpIX accumulated PDT pathways in cancer cells. The integration of these technologies provides promising prospects in the advancements of nanomolecule-driven theranostics for personalized therapy via targeted cancer imaging and therapy. Moreover, recently, a safety test was conducted utilizing ^111^In-labeled A_3_B-type Lactosome in brain metastasis model. The ^111^In-labeled A_3_B-type Lactosome showed selective accumulation in the brain metastases of the leptomeningeal, cerebral ventricle, and in bone metastasis while negligible distribution was observed in healthy brain, bone, and muscle tissues [55]. The ^111^In-labeled A_3_B-type Lactosome SPECT imaging contrast for head and neck metastasis was superior compared to ^201^TICI (Nihon Medi-Physics) for brain metastasis and ^99m^Tc-HMDP (Nihon Medi-Physics) for bone metastasis SPECT imaging and is therefore applicable as a diagnostic agent for the incurable meningeal dissemination [55]. In carefully preselecting patients, targeted nuclear therapies are found to be effective with a favorable safety profile.

However, despite the momentous progress utilizing Lactosome as a theranostic material, several challenges need to be addressed: 1. Practical considerations for clinical translation. To be translated clinically, any nanoplatforms involving the use of radionuclides involves local legal and ethical requirements. A team of highly specialized physicians and nurses together with radiochemists and medical physics experts will be required for patient-specific therapy planning including dosimetric monitoring and corresponding dose calculations to ensure the efficacy and safety of the treatment. 2. MSLN, overexpressed in a variety of malignancies, is a good target for anti-MSLN antibody-based diagnosis and therapy [18]. However, for clinical application, MSLN highly specific scFv is mandated as a positron emission tomography (PET) imaging reagent for rapid imaging and to avoid redundant immune responses. Yakushiji et al. has previously evaluated six antihuman MSLN-scFv-His-Tag clones (screened from a naïve phage library derived from human tonsil lymphocytes) in 14 different cancer cell lines and only one clone (H1a050) was found to have high reactivity to MSLN expressed cancer cells. This laborious prescreening effort is essential as a companion diagnosis strategy before the ^89^Zr-labeled scFv radiotheranostic concept can be realized. 3. The fluctuating dose requirements of therapeutic agents and imaging agents needs to be optimized to achieve optimal treatment results. 4. Lactosome-based theranostic platform integrates several moieties that drastically increased the complexity of this highly versatile nanoparticle. In addition, the variation in size and length of the poly (sarcosine) component complicates the up-scale production of Lactosome and that remains a great challenge for future manufacturers. Due to the high demand of iterative nanoplatforms, the robust and reproducible procedures that allow relatively easy and cost-effective scale-up of nano-theranostic platforms manufacturing is warranted. Hence, the design of simpler Lactosome through intelligible approaches is essential for future clinical evaluation and application. Up to now, pharmaceutical companies were investigating the possibilities of commercializing photodynamic therapy and photoacoustic diagnosing using the ICG-Lactosome [180,181], while the AB-type Lactosome is currently under consideration by licensing it to an overseas pharmaceutical manufacturer at a non-clinical level.

One of the major limitations of PDT in solid tumors is the penetration of the excitation light in tissue, where the fluence of the light decreases exponentially with the distance from the tissue surface. Hence, there is a growing interest in the development of remotely triggered theranostic nanoparticles called ‘smart theranostic’ platforms, which can concurrently diagnose a disease, start primary treatment, monitor response, and even, if required, initiate secondary treatments [13]. The remote triggering mechanisms include photodynamic, photothermal, photo-triggered chemotherapeutic release, electro-thermal, magneto-thermal, X-ray, ultrasound, and radiofrequency therapies. Specifically, a report regarding the modulation of a radioactive decay called Cherenkov luminescence (CL) via the use of nanoparticles has been highlighted. This CL offers the unique opportunity to modulate a radioactive decay signal for the first time via appropriately designed nanosensors, hence allowing the absorption and direct photoactivation by CL, and leading to a new class of smart, functional nanoparticles [179]. Thus, the “^89^Zr-labeled CPP-modified and photosensitizer (TPP)-loaded Lactosome” particles offer new avenues in biomedical research.

After refining the technical component of Lactosome technology, the integration of multi-modal delivery and targeting techniques utilizing magnetic targeting, ultrasound targeting, and even direct tissue-based targeting methods can be considered. Although more investigations are needed before Lactosome can be realized in the clinic, current findings indicate that Lactosome can potentially transform the approach of diagnosis and treatment to fulfill the possibility of a tailored, individualized nanomedicine. Once optimized for clinical translation, this platform promises to improve the quality of clinical care and treatments, which will ultimately save costs and improve disease burden, especially cancer.

## Figures and Tables

**Figure 1 life-11-00158-f001:**
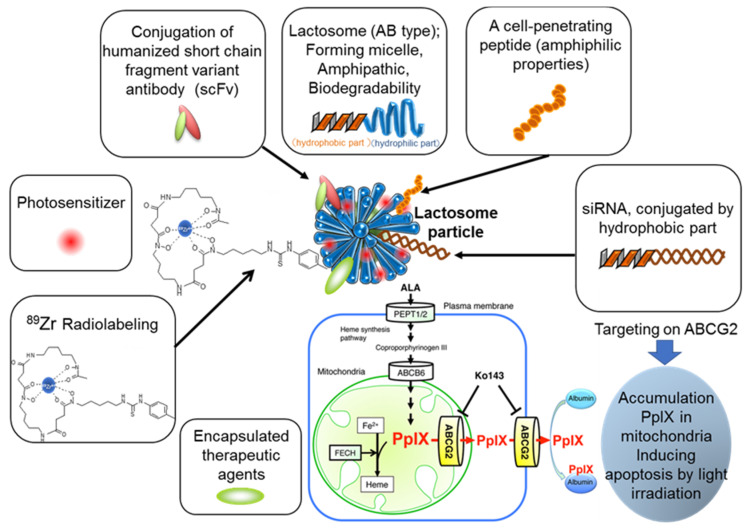
A novel theranostics technology with ^89^Zr, antibody variants, cell penetrating peptide (CPP), and siRNA modified Lactosome particles for siRNA knockdown and inducing apoptosis. PEPT1/2, oligopeptide transporters; ABCB6, ATP-binding cassette (ABC) transporter B6; ABCG2, ATP-binding cassette (ABC) transporter G2; FECH, ferrochelatase; PpIX, protoporphyrin IX; Ko143, specific inhibitor of ABCG2; ALA, 5-aminolevulinic acid.

**Figure 2 life-11-00158-f002:**
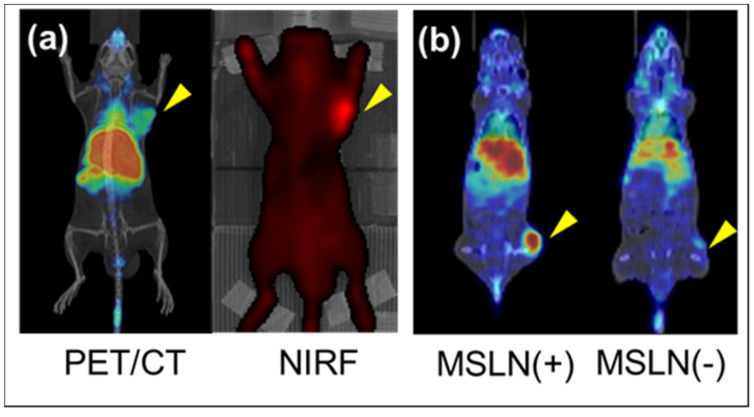
Biological distribution of ^89^Zr-labeled PET probes. (**a**) PET/CT and fluorescence images of mice bearing mesothelin (MSLN)-positive (NCl-N87) cancer cells 48 hr after administration of ^89^Zr- and Indocyanine Green (ICG)-labeled Lactosome. In all cases, the arrowhead signifies the transplanted tumors and (**b**) PET images of mice bearing tumors from both MSLN-positive (NCI-N87; MSLN(+)) and MSLN-negative (A431; MSLN(-)) cancer cells 48 hr after intravenous administration of ^89^Zr-labeled anti-MSLN-mAb (11–25).

**Figure 3 life-11-00158-f003:**
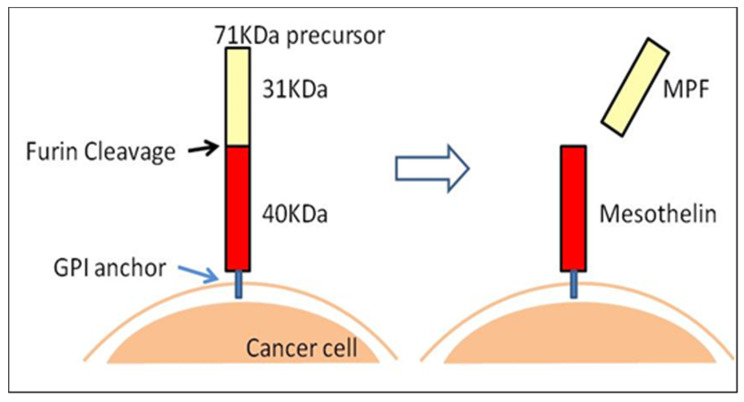
MSLN expression on cancer cells.

**Figure 4 life-11-00158-f004:**
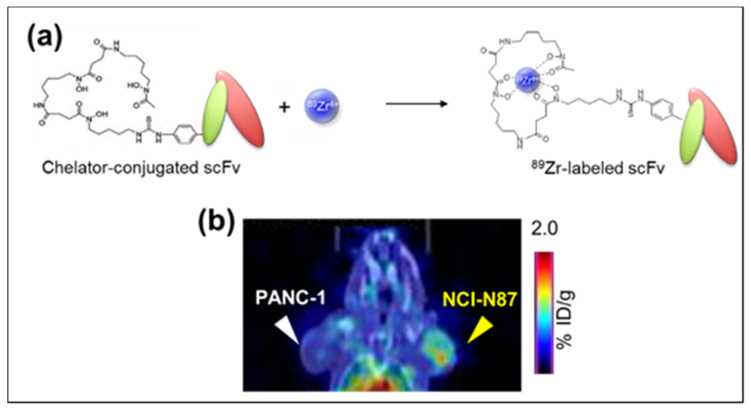
(**a**) Labeling of single chain Fv (scFv) of anti-MSLN IgG with deferoxamine and ^89^Zr and (**b**) PET/CT image of a mouse bearing tumors from both MSLN(+) and MSLN(-) cancer cells 3 hr after administration of ^89^Zr-labeled anti-MSLN-scFv. MSLN(+) gastric cancer cells (NCl-N87: right shoulder: yellow arrow) and MSLN (-) pancreatic cancer cells (PANC-1: left shoulder: white arrow) [85].

**Figure 5 life-11-00158-f005:**
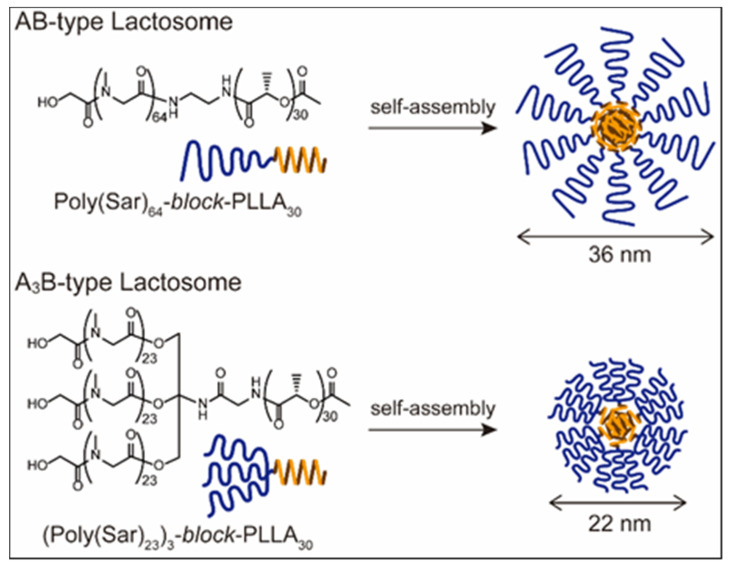
Illustration of the molecular structure of AB-type and A_3_B-type Lactosome [28].

**Figure 6 life-11-00158-f006:**
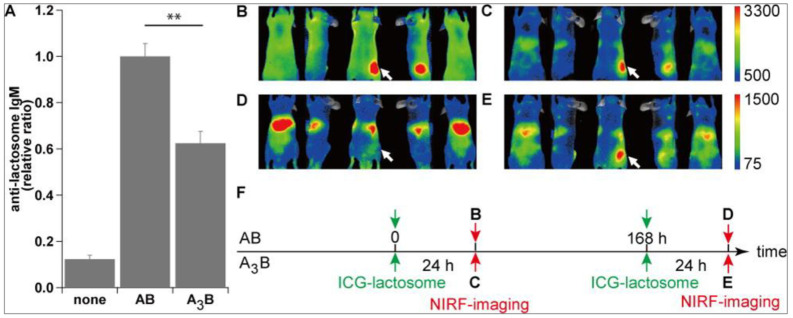
IgM productions at seven days after the administration of the AB- and A_3_B-type Lactosomes (**A**) (*n* = 3 per group). Anti-lactosome IgM productions are normalized by taking the anti-lactosome IgM production with the AB-type Lactosome as a reference, 1.0. Pharmacokinetic changes (NIRF images) upon multiple doses of the AB- (**B,D**) and A_3_B-type (**C,E**) Lactosome particles. The images D and E were taken at seven days after the first administration of the AB- and A_3_B-type Lactosome particles. The time schedule is shown in panel F. Reprinted (adapted) with permission from Hara, E., et al., Factors influencing in vivo disposition of polymeric micelles on multiple administrations. ACS Med Chem Lett, 2014. 5(8): p. 873-7. *Copyright © 2014, American Chemical Society*.

**Figure 7 life-11-00158-f007:**
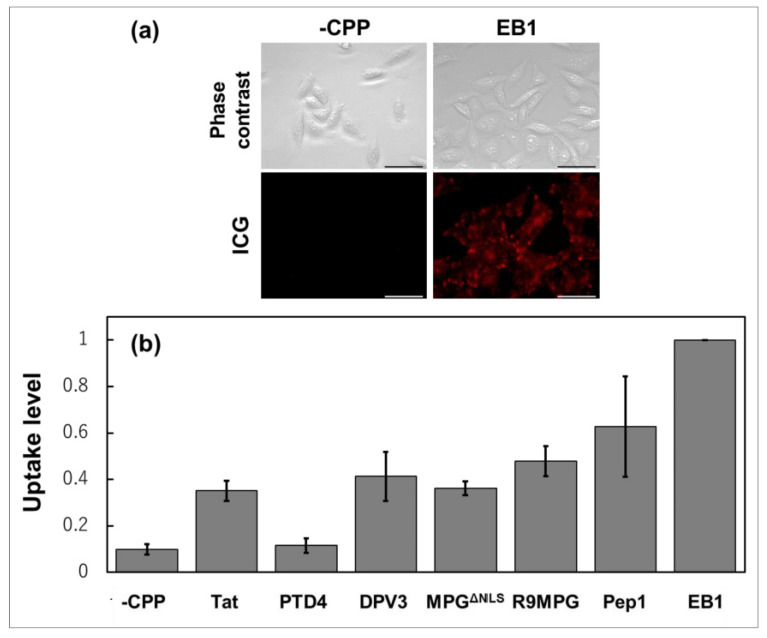
Cellular uptake of ICG-labeled CPP-modified Lactosome particles [31]. (**a**) Optical microscopy images of Chinese hamster ovary (CHO) cells cultured with ICG-labeled CPP-modified Lactosome particles. The scale bar is 20 μm. (**b**) Intracellular uptake level of CPP-modified lactosome particles (CPPs bearing a Cys residue was reacted with maleimide-PSar_56_-PLLA_30_ in the Lactosome particles).

**Figure 8 life-11-00158-f008:**
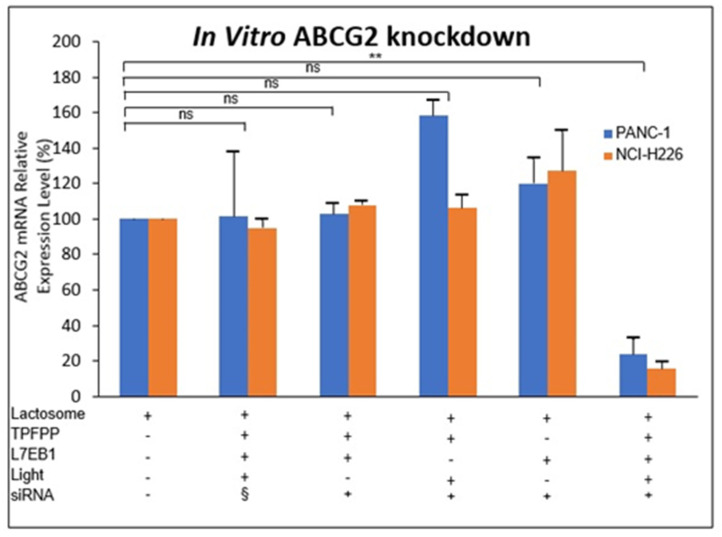
*In vitro* ABCG2 knockdown analysis of various formulations of Lactosome particles with ABCG2 siRNA incorporation on PANC-1 (MSLN(-)) and NCI-H226 (MSLN(+)) cancer cells. § signifies scrambled siRNA (*n* = 3). The p value: Not significant (ns) and *p* < 0.005(**). The siRNA was conjugated to the polymer (PLLA-SS-C_6_-siRNA) via the thiol modified sense strand added at the 6-carbon chain to the 5’end terminal while the L7EB1 and TPFPP photosensitizer was assembled into the hydrophobic core of Lactosome via hydrophobic interactions forming the siRNA loaded L7EB1/TPFPP-Lactosome particles collectively known as L7EB1/TPFPP/siRNA-Lactosome [153].

**Figure 9 life-11-00158-f009:**
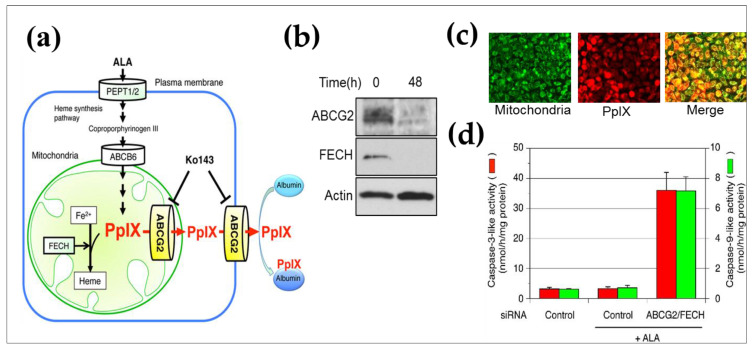
Aminolevulenic acid-photodynamic therapy (ALA-PDT). (**a**) Involvement of mitochondrial ABCG2 in the regulatory mechanism of ALA-mediated PpIX accumulation in malignant cells. (**b**) Western blot analysis of the knock down of ABCG2 and FECH by RNAi in HeLa cells, (**c**) fluorescence microscope analysis of mitochondria, and (**d**) ALA-mediated PpIX accumulation with photoirradiation leading to the activation of caspase-3 and caspase-9 like activity, which induces apoptosis. Cells with both ABCG2 and FECH knock down showed the highest apoptosis activity.

## Data Availability

Permission for reuse of Figure 6 was obtained from Hara, E., et al., Factors influencing in vivo disposition of polymeric micelles on multiple administrations. ACS Med Chem Lett, 2014. 5(8): p. 873-7. *Copyright © 2014, American Chemical Society* on Feb 12, 2021.

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
