# Peer review of "A Novel 89Zr-labeled DDS Device Utilizing Human IgG Variant (scFv): “Lactosome” Nanoparticle-Based Theranostics for PET Imaging and Targeted Therapy"

_life, 2021, doi:10.3390/life11020158_

Round 1
Reviewer 1 Report
This manuscript describes the comprehensive summary of Lactosome-based DDS. Introduction of scFv for targeting, PET probe for bioimaging, cell-penetrating peptide for internalization to the cells, photosensitizer for light-induced cell killing and photochemical internalization, siRNA and PpIX significantly improved the therapeutic effects by Lactosomes. The authors well-summarized the effects of each component of functional Lactosomes on their biodistribution, immune responses, cellular internalization and therapeutic effects. Therefore, current manuscript would provide useful information to overview the molecular design of polymer-based nanomedicine and its biomedical application. This manuscript can be acceptable after minor corrections as listed:
-Figure 1: “siRNA, conjugated by hydrophobic part” would be changed to “siRNA, conjugated by hydrophilic part”.
-Comparison data between Lactosomes and representative nanomedicine on biodistribution or therapeutic effects should be presented as Figures to show the superiority of Lactosomes clearly.
Author Response
Response) Thank you for your valuable comments.
- Figure 1: “siRNA, conjugated by hydrophobic part” would be changed to “siRNA, conjugated by hydrophilic part”.
Several modifications were previously attempted and the most efficient siRNA conjugation is via the hydrophobic PLLA modification. The siRNA illustration, conjugated by hydrophobic part was changed. The method for hydrophobic modified siRNA via disulfide exchange can be found in this reference;
Lim, M.S.H.; Nishiyama, Y.; Ohtsuki, T.; Watanabe, K.; Kobuchi, H.; Kobayashi, K.; Matsuura, E. Lactosome-conjugated siRNA nanoparticles for photo-enhanced gene silencing in cancer cells. Journal of pharmaceutical sciences 2021, https://doi.org/10.1016/j.xphs.2021.01.026, doi:https://doi.org/10.1016/j.xphs.2021.01.026
We have also changed the illustration on Figure 1 at Page 5.
-Comparison data between Lactosomes and representative nanomedicine on biodistribution or therapeutic effects should be presented as Figures to show the superiority of Lactosomes clearly.
Response) We have added [Figure 6. IgM productions at 7 days after the administration of the AB- and A3B-type Lactosomes (A) (n = 3 per group). Anti-lactosome IgM productions are normalized by taking the anti-lactosome IgM production with the AB-type Lactosome as a reference, 1.0. Pharmacokinetic changes (NIRF images) upon multiple doses of the AB- (B,D) and A3B-type (C,E) Lactsosome particles. The images D and E were taken at 7 days after the first administration of the AB- and A3B-type Lactosome particles. The time schedule is shown in panel F. Reprinted (adapted) with permission from Hara, E., et al., Factors influencing in vivo disposition of polymeric micelles on multiple administrations. ACS Med Chem Lett, 2014. 5(8): p. 873-7. Copyright © 2014, American Chemical Society.] on Page 14 which compares the superiority of A3B-type Lactosome compared to AB-type Lactosome.
Unfortunately, as of now, we do not have head to head comparison of Lactosome with other representative nanomedicine such as Liposomes or PEG Liposomes. However, we will consider incorporating such experiments for future studies.
Reviewer 2 Report
Dear Authors,
Thank you for your contribution to MDPI Life. Your review article is well written and clearly lays out the potential advantages of 89-Zr-labeled lactosomes for PET imaging. I felt adequate treatment of the diagnostic significance of these lactosomes was adequately discussed. I consider your article acceptable in its current form. However, addressing a few considerations listed below would strengthen the article and perhaps broaden its readership. Best of luck with your future endeavors.
-- It would be nice to see a deeper discussion of what specific barriers must be overcome to translate this technology to humans.
--What manufacturing concerns exist for scaling these technologies to clinical scales?
-- Could you comment on whether or not there has been any commercial / industrial interest in this device?
-- Can you include a section discussing multi-modal delivery and targeting techniques, if any have been devised for these lactosomes? For example, magnetic targeting, ultrasound targeting, or tissue based targeting methods?
Author Response
Thanks for your valuable comments.
- It would be nice to see a deeper discussion of what specific barriers must be overcome to translate this technology to humans.
Response) We have added extra discussion of Page 20, Line 843-848, and Page 21, Line 849-850 as follows,
“Moreover, recently, a safety test was conducted utilizing 111In-labeled A3B-type Lactosome in brain metastasis model. The 111In-labeled A3B-type Lactosome showed selective accumulation in the brain metastases of the leptomeningeal, cerebral ventricle and in bone metastasis while negligible distribution was observed in healthy brain, bone, and muscle tissues[55]. The 111In-labeled A3B-type Lactosome SPECT imaging contrast for head and neck metastasis was superior compared to 201TICI and 99mTc-HMDP SPECT imaging and is therefore applicable as a diagnostic agent for the incurable meningeal dissemination[55].”
And Page 21 Line 867-901 as follows,
However, despite the momentous progress utilizing Lactosome as a theranostic material, several challenges need to be addressed; 1. Practical considerations for clinical translation. To be translated clinically, any nanoplatforms involving the use of radionuclides involves local legal and ethical requirements. A team of highly trained physicians and nurses together with radiochemists and medical physics experts will be required for patient-specific therapy planning including dosimetric monitoring and corresponding dose calculations to ensure the efficacy and safety of the treatment. 2. MSLN, overexpressed in a variety of malignancies, is a good target for anti-MSLN antibodybased diagnosis and therapy[18]. However, for clinical application, MSLN highly specific scFv is mandated as a positron emission tomography (PET) imaging reagent for rapid imaging and to avoid redundant immune responses. Yakushiji et al. has previously evaluated six antihuman MSLN-scFv-His-Tag clones (screened from a naïve phage library derived from human tonsil lymphocytes) in 14 different cancer cell lines and only one clone (H1a050) was found to have high reactivity to MSLN expressed cancer cells. This laborious pre-screening effort is essential as a companion diagnosis strategy before the 89Zr-labeled scFv radiotheranostic can be realized. 3. The fluctuating dose requirements of therapeutic agents and imaging agents needs to be optimized to achieve optimal treatment results. 4. Lactosome-based theranostic platform integrates several moieties which drastically increased the complexity of this highly versatile nanoparticle. In addition, the variation in size and length of the poly(sarcosine) component complicates the up-scale production of Lactosome and that remains a great challenge for future manufacturers. Due to the high demand of iterative nanoplatforms, the robust and reproducible procedures which allows relatively easy and cost-effective scale-up of nanotheranostic platforms manufacturing is warranted. Hence, the design of simpler Lactosome through intelligible approaches is essential for future clinical evaluation and application. Up to now, pharmaceutical companies were investigating the possibilities of commercializing photodynamic therapy and photoacoustic diagnosing using the ICG-Lactosome[180, 181], while the AB-type Lactosome is under consideration by licensing it to and overseas pharmaceutical manufacturer at a nonclinical level. After refining the technical component of Lactosome technology, the integration of multi-modal delivery and targeting techniques utilizing magnetic targeting, ultrasound targeting and even direct tissue-based targeting methods can be considered. Although more investigations are needed before Lactosome can be realized in the clinic, current research indicates that Lactosome can potentially transform the approach of diagnosis and treatment to fulfill the possibility of a tailored, individualized nanomedicine.
- What manufacturing concerns exist for scaling these technologies to clinical scales?
Response) We have added extra discussion of Page 21, Line 882-891 as follows,
“3. The fluctuating dose requirements of therapeutic agents and imaging agents needs to be optimized to achieve optimal treatment results. 4. Lactosome-based theranostic platform integrates several moieties which drastically increased the complexity of this highly versatile nanoparticle. In addition, the variation in size and length of the poly(sarcosine) component complicates the up-scale production of Lactosome and that remains a great challenge for future manufacturers. Due to the high demand of iterative nanoplatforms, the robust and reproducible procedures which allows relatively easy and cost-effective scale-up of nanotheranostic platforms manufacturing is warranted. Hence, the design of simpler Lactosome through intelligible approaches is essential for future clinical evaluation and application.”
- Could you comment on whether or not there has been any commercial / industrial interest in this device?
Response) We have added extra discussion of Page 21, Line 891-895 as follows,
“Up to now, pharmaceutical companies were investigating the possibilities of commercializing photodynamic therapy and photoacoustic diagnosing using the ICG-Lactosome[180, 181], while the AB-type Lactosome is under consideration by licensing it to and overseas pharmaceutical manufacturer at a non-clinical level.”
-- Can you include a section discussing multi-modal delivery and targeting techniques, if any have been devised for these lactosomes? For example, magnetic targeting, ultrasound targeting, or tissue based targeting methods?
Response) Regarding the use of multi-modal delivery and targeting techniques, we aim to establish these components with Lactosome once we have thoroughly scrutinize and establish the technical issues regarding Lactosome particles. We have added extra discussion of Page 21, Line 896-898 as follows,
“After refining the technical component of Lactosome technology, the integration of multi-modal delivery and targeting techniques utilizing magnetic targeting, ultrasound targeting and even direct tissue-based targeting methods can be considered.”